# A Generic Method for Fast and Sensitive Detection of Adeno-Associated Viruses Using Modified AAV Receptor Recombinant Proteins

**DOI:** 10.3390/molecules24213973

**Published:** 2019-11-03

**Authors:** Mengtian Cui, Yabin Lu, Can Tang, Ran Zhang, Jing Wang, Yang Si, Shan Cheng, Wei Ding

**Affiliations:** 1Department of Genetics and Developmental Biology, School of Basic Medical Sciences, Capital Medical University, Beijing 100069, China; cmtecho@163.com (M.C.); luyabin@ccmu.edu.cn (Y.L.); jingwang@ccmu.edu.cn (J.W.); 13601333334@163.com (Y.S.); weiding@ccmu.edu.cn (W.D.); 2MOE Laboratory of Protein Science and Collaborative Innovation Center of Biotherapy, School of Medicine, Tsinghua University, Beijing 10084, China; zhangr16@mails.tsinghua.edu.cn; 3Beijing Key Laboratory of Cancer & Metastasis Research, Capital Medical University, Beijing 100069, China

**Keywords:** AAV, AAV receptor, virus detection, immunosorbent assay, gene therapy

## Abstract

Adeno-Associated Viruses (AAV) are widely used gene-therapy vectors for both clinical applications and laboratory investigations. The titering of different AAV preparations is important for quality control purposes, as well as in comparative studies. However, currently available methods are limited in their ability to detect various serotypes with sensitivity and convenience. Here, we took advantage of a newly discovered AAV receptor protein with high affinity to multiple AAV serotypes, and developed an ELISA-like method named “VIRELISA” (virus receptor-linked immunosorbent assay) by adopting fusion with a streptavidin-binding peptide (SBP). It was demonstrated that optimized VIRELISA assays exhibited satisfactory performance for the titering of AAV2. The linear range of AAV2 was 1 × 10^5^ v.g. to 5 × 10^9^ v.g., with an LOD (limit of detection) of 5 × 10^4^ v.g. Testing of VIRELISA for the quantification of AAV1 was also successful. Our study indicated that a generic protocol for the quantification of different serotypes of AAVs was feasible, reliable and cost-efficient. The applications of VIRELISA will not only be of benefit to laboratory research due to its simplicity, but could also potentially be used for monitoring the circulation AAV loads both in clinical trials and in wild type infection of a given AAV serotype.

## 1. Introduction

Adeno-Associated Virus (AAV) is a small virus with a single-stranded DNA genome encapsidated in an icosahedral protein capsid shell. AAV has been studied for more than half of a century and has been used as a common vector for gene therapy for more than 30 years [1]. AAV delivery of therapeutic genes has been successfully applied in the treatment of several monogenic diseases, as well as in the early phase of clinical trials for treating α1 antitryptase deficiency, thalassemia B, hemophilia A, and hemophilia B [1,2]. There are currently more than 20 open clinical trials making use of AAV vectors of multiple serotypes for the treatment of generic and acquired diseases, including US FDA-approved ones for the delivery of RPE65 gene to viable retinal cells in patients with visionary problems [2,3]. The advantages of of AAV-based vectors include their demonstrated safety due to their non-pathogenecity and their ability to maintain long-term gene expression [4,5]. Although significant improvements for recombinant AAV (rAAV) production and purification have been reported [1,6,7], methods for monitoring the quality or existence of rAAV virions remain insufficient, and this is a constant concern in clinical-grade rAAV manufacture and reliable GMP-compliant vector production for scalability [8,9].

Although AAV2 remains the dominant subtype of AAVs used in clinical protocols, many other AAV serotypes have started to enter trial phases following development and optimization at research laboratories, including AAV1, AAV8, AAV9, AAV5, etc. [10,11,12,13]. The classical methods for the purification of AAVs is through CsCl gradient ultracentrifugation, which potentially enables the physical separation of full virus particles from empty particles (only viral capsid without encapsulated viral DNA) [14]. Recently, several protocols using ion exchange or affinity chromatography have been developed, demonstrating advantages in terms of both efficiency and scalability [15,16]. Quantitative polymerase chain reaction (qPCR) is a conventional method for titering AAV viral genome (v.g.) copy numbers, substituting for traditional slot blot using isotope probes for hybridization [17]. As is commonly recognized, the quality of different AAV preparation batches could vary to a certain extent as a result of the purification methods used or the effects of differences among the viral packaged genes with respect to length and sequence. One major influencing factor is the percentage of the empty virions with no transduction activities [1,18]. PCR-based AAV quantification methods are not able to provide information for the estimation of loss of integrity in viral particles. Furthermore, PCR assays could easily be compromised by salts or other contaminants, including recombinant plasmid DNAs or Rep proteins from insufficient digestion, leading to inaccurate measurements [1,18]. Therefore, an accurate, fast, and higher-throughput method for the titering of AAV particles is necessary and essential, and still in demand for continuous improvements.

The human gene *KIAA0319L* was cloned in 2001 from a size-fractionated fetal brain cDNA library, which encodes a protein of 639 amino acid residuals containing a C6 region near the N terminus, a MANSC domain and five polycystic kidney disease (PKD) domains [19]. In 2016, KIAA0319L was determined to be a molecule that could serve as a universal AAV receptor (AAVR). In AAVR knock-out cells, AAV infection was significantly reduced in different rAAV serotypes, including AAV1, AAV2, AAV3b, AAV5, AAV6, AAV8, and AAV9, which were suggested to bind AAVR through distinct PKD domains. PKD2 is the major interface interacting with the AAV2 capsid, but not AAV4 or possibly other serotypes [20,21]. This indicates that other PKD domains are not only important for the maintenance of the spatial structure of AAVR, but that it may also contribute directly to the high-affinity binding of certain AAV subtypes [22,23]. It has also been suggested that AAVR could potentially be subjected to glycosylation and form various cellular glycoproteins; however, it is the N-linked glycosylation that was indicated not to be required for AAV2 binding in order to enhance viral transduction [24]. This made it feasible to develop methods using prokaryotic expression of AAVR recombinant proteins for the detection of AAV virions in in vitro assays. 

Considering the factors that define a good immunosorbent detection system (including the specificity, sensitivity and costs, etc.), wherein the target protein is to be identified from the crude extract or complex solution samples, both His-fusion and His-SBP-fusion AAVR proteins were used for the development of VIRELISA (Virus Receptor linked immunosorbent assay). SBP (streptavidin binding peptide) is a small tag of 38 amino acids that binds streptavidin with high affinity (Kd~2.5 nM). The engaged binding of SBP tagged AAVR to HRP-SAV (HRP-Streptavidin) made possible a colorimetric detection method that can be directly visualized. Nevertheless, the biotin ligand can still be used to compete off SBP in complex with SAV, which leaves further opportunity to expand the use of SBP-fusion proteins for additional verification purposes [25,26,27]. By adding His tags to AAVR or AAVR-SBP, the purification and characterization of the prokaryotic expressed recombinant proteins can be performed with ease. VIRELISA is designed to be a generic method for the quantification of different serotypes of AAVs (Figure 1). As demonstrated in the present study, a primary application could be in the titration of AAV2 and AAV1 particles from the fractions of CsCl gradient centrifugation during the virus purification procedures. The method could be further implemented in other situations to titrate AAV particles for purposes of both laboratory research and clinical application.

## 2. Results

### 2.1. AAVR Facilitated AAV2 Transportation to Trans-Golgi and Enhanced AAV2 Transduction through its High-Affinity Binding to the Viral Capsid

As a normal functional cellular protein, the localization of AAVR is naturally concentrated around the trans-Golgi network (TGN), as shown by the strong colocalization from the immunofluorescent staining of AAVR and the TGN46 marker (Figure 2A). Using Alexa 568-labeled AAV2, we observed that infected AAV2 in HeLa cells at an m.o.i. of 2000 v.g./cell was also localized to TGN, and the fluorescent signal overlapped with the staining of AAVR. This suggested the binding of AAV2 with its receptor could also be an important requirement for viral trafficking, in addition to the binding at the cell surface. By overexpressing the wild type AAVR following plasmid transfection in HeLa cells (Figure 2B), the luciferase activity from the reporter transgene expression was significantly increased by 2~2.5 folds, demonstrating that binding with AAV2 facilitated the viral transduction of the infected host cells (Figure 2C) .

### 2.2. Preparation and Characterization of Recombinant AAVR Proteins

Removing the single-pass membrane-spanning helix and the downstream intracellular region will presumably increase the solubility of recombinant AAVR proteins. In this study, we constructed the prokaryotic expressing constructs of AAVR fragments containing the extracellular PKD1-5, which is known to be sufficient for AAV binding, and fused the sequence with either the His or AAVR SBP-His tag. From the transformed *E. coli.* (BL21) (Figure 3A), the recombinant fusion proteins were purified by steps of nickel affinity chromatography and additional gel filtration chromatography (Figure 3B). The obtained products were subjected to SDS-PAGE and stained by Coomassie brilliant blue for purity and integrity examinations. The concentrations of the purified AAVR-SBP-His and AAVR-His protein were determined to be 1.2 and 0.8 mg/mL, respectively, by BCA assays. To determine the binding affinity between AAV1 and AAVR, a BIAcore assay was performed. The measured Kd was about (255 nM), which is a substantially high number that indicates the strong association of the formed complex (Figure 3C). To further test whether the obtained proteins could be applied for viral detection purposes, we performed virus overlay experiments following the procedures shown in Figure 4A. Compared to A20, a well-characterized antibody to intact AAV2 particles, our recombinant proteins were able to produce similar results in terms of both sensitivity and specificity for the detection of AAV2 virions from different titers (Figure 4B).

### 2.3. Refinement of the “VIRELISA” Assay Conditions

Following the principle of ELISA, we started to assemble an assay named “VIRELISA” as a method for the detection of AAVs. The concentration of the immunosorbent agent (the capture antibody) was a most crucial factor to be decided, as this could significantly affect the sensitivity, reproducibility, and stability of the assay. We screened AAVR-SBP-His concentrations (50 μg/mL, 100 μg/mL) in combinations HRP-SAV dilutions (1:1000, 1:2000, 1:5000) and used positive-to-negative (P/N) ratios for evaluations. The average absorbance of OD450 from sample duplicates (P-positive) or buffer controls (N-negative) was used to calculate the P/N ratios. As was showed in Table 1, a significantly improved P/N ratio (3.4) was observed in the condition of 50 μg/mL of AAVR-SBP-His for detecting AAV2 (in amounts of 1.0 × 10^8^ v.g.). Similarly, optimized HRP-SAV dilution (1:5000) and incubation conditions with ABTS (15 min at 37 °C) were determined based on the P/N ratios (Table 1).

### 2.4. Method Validation for “VIRELISA”

#### 2.4.1. Calibration Curve, LOD and LOQ

A standard curve for AAV2 detection was obtained with seven concentrations of different dilutions. The correlation in the standard curve (Y = 0.0254 ln(X) + 0.1458, R^2^ = 0.9900) was in a linear range of the absorbance (450 nm) between 0.425 and 0.702 and the virus concentration of 1 × 10^5^ v.g. to 5 × 10^9^ v.g. in the test volume of 100 μL. The limit of detection (LOD) and the limit of quantification (LOQ) values were determined to be 5 × 10^4^ and 1 × 10^5^, respectively. The upper limit of quantification (ULOQ) value was calculated to be 5 × 10^9^ v.g.

#### 2.4.2. Precision

To demonstrate and ascertain the robustness and precision of the VIRELISA assay, several duplicate tests of the AAV2 fractions purified from CsCl gradient centrifugation were assessed, ranging from 1.0 × 10^6^ v.g. to 1.0 × 10^8^ v.g. As shown in Table 2, the intra-assay RSD% (relative standard deviation) ranged from 3.51% to 4.05%. The inter-assay RSD% ranged from 3.92% to 4.59%.

#### 2.4.3. Recovery

Spike-in experiments were performed to determine the recovery index of the VIRELISA assay. We first randomly selected a fraction from the CsCl separation of the AAV2 preparation, and used VIRELISA to determine the viral content as 1.0 × 10^6^ v.g. in total. In addition, purified AAV2 of 1.0 × 10^6^ v.g. to 1.0 × 10^8^ v.g. was then added to the AAV2 fraction to produce a sample series. Evaluations were performed by calculating the ratio between measured and expected values of AAV2 particles. The percentage recovery of VIRELISA assay in AAV2 particle numbers was between 102.2% and 108.4% (Table 3), indicating that recovery of the fraction from CsCl gradient purification was quantitative, and that AAV2 particles were accurately measured in the real sample.

#### 2.4.4. Selectivity

In order to determine the selectivity of VIRELISA, boiled AAV2 in denatured fragments or interference proteins (BSA in our case) was added into the purified AAV2 samples with known v.g. in titers, and subjected to re-examination by VIRELISA. The effect of sodium deoxycholate (DEO) or CsCl contaminants was also assessed. The recoveries of AAV2 in these samples were calculated from the VIRELISA titers of AAV2 determined before and after addition of interfering substances. The results in Table 4 showed that no major cross-reactivity was identified with the addition of interference compounds.

### 2.5. VIRELISA Assay for Different AAV Serotypes

Since AAVR was reported to be able to bind various AAV serotypes, we also explored the potential of our VIRELISA assay to be used for titering other serotypes of AAVs from laboratory preparations. Table 5 summarizes the VIRELISA results for AAV1 detection with the comparison to qPCR assays. It was found that the VIRELISA assay was capable of titering AAV1 with sufficient accuracy and performed comparably in terms of both precision and recovery to conventional qPCR-based assays.

## 3. Discussion

ELISA is a high-sensitivity homogeneous immunoassay that has been used for the detection of a great variety of antigens for the purpose of biomedical research and clinical examination for decades. Among the ELISA kits that have been developed for AAV serotypes, monoclonal antibody A20 is the most successful reagent, and it has been widely used for assaying AAV2, the most prevalent AAV serotype in humans [28,29,30,31]. Unlike many viral antibodies, which only recognize the epitope of viral proteins, A20 binds the capsid of intact viral particles, which is a desirable property, as it makes it possible to directly indicate the infectious capability of the detected virus samples [24]. However, A20 is only applicable for the serotypes AAV2 and AAV3 [28,31]. As other AAV serotypes have been developed into successful gene therapy vectors and have demonstrated certain advantages, there is a growing need for similar detection methods. Unfortunately, such products are still rare or rather expensive. The identification of the universal high-affinity receptor of AAVs (AAVR) provided a good opportunity for the design and development of the required method [20]. The icosahedral structure of AAV supplies multiple spike regions that can be captured by AAVR. The most recent structure of the AAV2–AAVR complex obtained using cryo-electron microscopy revealed that the PKD2 domain of AAVR bound directly to the spike region of the AAV2 capsid adjacent to the icosahedral three-fold axis [22]. This suggests the possibility of using AAVR fragments as a serotype-specific probe to mimic the binding of A20, while targeting a different loci. However, since the wild type AAVR in full length is capable of binding different AAV serotypes with high affinity, as shown from the BIAcore analysis of the AAV–AAVR interaction (Figure 3C), we intended to develop a generic ELISA-based titration method with broader applications for the detection of various AAV particles. Therefore, recombinant His-tagged AAVR and AAVR-SBP-His proteins were produced as capture and probe reagents, respectively. 

The titering of the AAV preparation aims to evaluate its infectious activity and estimate the m.o.i. to be used. The methods currently being used are predominantly derived from quantitative PCR protocols [32,33]. It was recognized that different batches of AAV preparation were contaminated with variable percentages of empty virions, which could potentially significantly influence the functional infection dosage. Comprehensive analysis of the VIRELISA results with the classical qPCR titer could, in theory, resolve such problems. A critical condition to be achieved is that VIRELISA should have a performance equal to the PCR-based titering methods in terms of both sensitivity and accuracy. Based on the present study, VIRELISA was demonstrated to possess the ability to run large numbers of samples using a small sample volume in a homogeneous environment, exhibiting satisfactory sensitivity and a dynamic working range. The detection limit was 5 × 10^4^ v.g., which allows the measurement of low concentrations of AAV particles. The assay was found to be robust to interference from chemical reagents, and this was superior to PCR reactions, which are frequently compromised under certain conditions. Under the normal conditions of AAV titering applications, the precision profiles of VIRELISA and qPCR were similar and produced comparable results. However, the influencing factors for VIRELISA and qPCR could be different, and the assay that is able to give the results that best represent the functional viral titers in transduction experiments remains to be tested in the future. 

In addition to the advantages of VIRELISA in detecting multiple AAV serotypes, as demonstrated for AAV1 titration in the present study, VIRELISA can potentially be used for analyzing different types of samples, as well. It is worthwhile further expanding the application of VIRELISA to detect the plasma load of administered AAV in patients or animal models subjected to AAV-based gene therapy protocols; or it could potentially also be tried for evaluating different wild type AAV infections in individuals. In addition, the SBP tagging construct produced from this study is an important development for expanding the application of AAVR proteins. SBP-AAVR alone could be an ideal reagent for probing AAV virions in virus overlay assays (Figure 4B). It not only improved the performance of the VIRELISA kit, enabling a colorimetric qualitative estimation of AAV existence without the requirement of any instruments, but could also be used for the affinity purification of AAV particles of different serotypes as well, where commercially available streptavidin matrix can be selected. By design, the binding constant of SBP to avidin is in between the AAV binding to receptors and avidin binding to biotin, which in principle could make it versatile in practical application. Moreover, with certain strategies for making chimeric AAV capsid, like the four plasmid mixing method [34,35], AAVR is can theoretically be perceived to be binding the produced virions, and therefore could be detected (albeit qualitatively) using this method. 

## 4. Materials and Methods

### 4.1. AAV Production and Purification

Triple plasmid transfection with polyethylenimine (PEI) was used to produce recombinant AAV2 or AAV1. HEK293T cells were seeded in 150-mm plates and cultured until they reached 80% confluence in Dulbecco’s modified essential medium (DMEM, Corning, Corning, NY, USA) containing 10% fetal bovine serum (ExCell Bio, Washburn, MO, USA ) and 1% penicillin–streptomycin antibiotics (Gibco, Billings, MT, USA) at 37 °C. The cells were co-transfected with plasmids of pRC (code for Rep and Cap proteins) (10 μg), pHelper (20 μg) and 10 μg of either pAV2-luc or pAV1-luc. At 72 h post transfection, cells were collected by centrifugation at 1000 ×*g* at 4 °C for 30 min, and then resuspended in buffer containing 10 mM Tris-HCl, pH 8.0. Four rounds of freeze–thaw cycles were carried out by alternating baths of dry ice/ethanol and 37 °C water. Digestion with DNase I (200 units in 1.5 mL of 1% sodium deoxycholate with additional 0.05% trypsin) for 1 h at 37 °C was performed following sonication of the samples. The AAV crude lysate was prepared by centrifugation at 10,000× *g* for 10 min at 4 °C from the collected supernatant. 

The AAV crude lysate was diluted with 10 mM Tris-HCl (pH 8.0) to a final volume of 3.9 mL and then bottom-loaded to a discontinuous gradient of CsCl (1.25 g/cm^3^) and CsCl (1.50 g/cm^3^) in an 8.9 mL ultracentrifuge tube (Beckman Coulter, Indianapolis, Indiana, USA). Following ultracentrifugation at 200,000× *g* at 4 °C for 8 h, fractions of 500 μL each were collected. The virus-containing fractions were combined and subjected to desalt using a Ultrafiltration kit of 3 kD Cutoff (10 m; Millipore, Burlington, MA, USA). The purified AAV was characterized by silver staining and qPCR before being stored at −80 °C. 

### 4.2. Quantification of Viral DNA by qPCR

The packaged viral DNA from AAV particles was extracted by a genomic nucleic acid preparation kit (TaKaRa, Dalian, China). The PCR primers 5’-GGAACCCCTAGTGATGGAGTT-3’ and 5’-CGGCCTCAGTGAGCGA-3’ were used to amplify a 106 bp fragment near the pAV2 ITR region. SYBR Premix Ex Taq™ II (TaKaRa) was used for qRT-PCR reactions. A linear standard curve with a coefficient above 0.99 was used for quantification of the PCR amplification efficiency between 95% and 105%. Samples were run in triplicate, and the average of the calculated copy numbers was used to represent the titer of the obtained AAVs. 

### 4.3. Preparation of Recombinant AAVR Proteins

The AAVR complementary DNA encoding PKD domains 1–5 was fused at the C-terminus with a 6× His-tag or streptavidin binding peptide (SBP) and 6× His-tag in tandem and cloned into a pET-28a vector. The amino acid sequences of AAVR PKD domains 1–5 and the SBP tag, as well as the primers used for the cloning of PET28a-AAVR-SBP-His plasmid, are supplied in the Appendix A. The fusion protein was isolated from the crude lysate of transformed *E. coli.* by nickel affinity chromatography (Qiagen, Venlo, Netherlan) and further purified by additional gel filtration chromatography using a Superdex-200 column (GE Healthcare, Chicago, IL, USA). The wild type AAVR fragment of the soluble extracellular region was prepared similar to the method previously described [20]. The obtained proteins were analyzed by sodium dodecyl sulfate polyacrylamide gel electropheresis (SDS-PAGE) and stained by Coomassie blue staining. The protein concentrations were determined with a BCA™ Protein Assay Kit (ThermoFisher, Waltham, MA, USA). 

### 4.4. Blotting for Intact AAV Viral Particles (Overlay Assay)

AAV dilutions from (total 1 × 10^9^ v.g.) were loaded onto nitrocellulose filters (0.42 μm, GE Healthcare) at room temperature with a vacuum slot apparatus. The filters were blocked with 2% BSA in PBST for 1 h, and then probed with AAVR-SBP-His proteins by incubation for 1 h. After being washed six times at 5 min intervals, the filter was incubated with HRP-conjugated streptavidin (HRP-SAV) (Solarbio, Beijing, China) for 60 min. The blots were visualized by the enhanced chemiluminescence method according to the instructions from Amersham (GE Healthcare Biosciences, Piscataway, NJ, USA). For comparison controls, monoclonal antibody A20 was used for the parallel procedures, where incubation with HRP-conjugated goat anti-mouse IgG (1:10,000 dilution in PBS supplemented with 1% non-fat milk) was performed. PBST was used as the negative control sample.

### 4.5. “VIRELISA” (Virus Receptor-Linked Immunosorbent Assay)

#### 4.5.1. Assay Procedures

Diagrams of the sandwich VIRELISA assay are shown in Figure 1. Each well of the 96-well plates was precoated with AAVR-His proteins by adding 10 μg purified AAVR-His proteins diluted in 100 μL PBST and dried at room temperature. The AAV samples in 100 μL volume were added into the coated well and incubated at RT for 1 h. After being washed with PBST, the plates were blocked for 1 h with 100 μL BSA (10 μg/mL). Then, amounts of 100 μL of AAVR-SBP-His proteins (final concentration 10 μg/mL) in PBST were added and incubated for 1 h, HRP-conjugated streptavidin (HRP-SAV) in 100 μL PBST (final concentration 10 mmol) was subsequently added after extensive washing. The final reaction for 15 min at 37 °C was performed by adding 0.5 mg/mL ABTS (Sigma-Aldrich) and 0.01% hydrogen peroxide into the wells. A BioTek ELx 800 TS Absorbance Reader (BioTek, Winooski, VT, USA) was used to scan the plates at 450 nm. 

#### 4.5.2. LOD and LOQs

For method validation of VIRELISA, assay calibration was first performed using a series of high-purity AAV2 dilutions from 1.0 × 10^5^ to 5.0 × 10^9^ v.g. in PBS from a stock of 1.0 × 10^12^ v.g./mL. The limit of detection (LOD), which represents the virions required to give a signal equal to the background (blank) plus three times the standard deviation of the blank, the limit of quantification (LOQ), including the lower LOQ (LLOQ) as the higher value between doubled level of the LOD and the point where the CV falls below 20%, as well as the upper LOQ (ULOQ) as the point at which the calculated precision does not exceed 15% of the CV and the accuracy is within 15% of the expected concentration, were determined through the designed experiments.

#### 4.5.3. Precision, Recovery and Selectivity

The results of the intra-assay and inter-assay comparisons were used to estimate the precision of VIRELISA using AAV2 samples from 1.0 × 10^8^ to 1.0 × 10^6^ v.g. Triplicates of each sample were placed randomly on the same plate and the intra-assay variations from six determinations were analyzed. For the inter-assay tests, the results of the same sample in three independent tests (one duplicate per day, and each sample in a different position on the plate in each test) were summarized. The rate of recovery was assessed by adding 50 μL AAV2 from 1.0 × 10^6^ to 1.0 × 10^8^ v.g. to the 1.0 × 10^8^ v.g. purified AAV2 sample in 50 μL. The ratio between the measured and expected values from a matrix of spiked samples was calculated for estimation. The selectivity was tested by adding boiled AAV2, DEO or BSA as interference materials to the fractionated samples from CsCl gradient purification of AAV lysate. The cross-reactivity was expressed as the percent ratio between measured and expected values.

### 4.6. AAV Labeling and Fluorescence Microscopy

The purified AAV2-Luc capsid was labeled by Alexa Fluor™ 568 Protein Labeling Kit (ThermoFisher, Waltham, Massachusetts, USA). Viral stock was incubated in 50 nmol/L reactive Alexa568 dye for 1 h in 37 °C. The reaction mixture was loaded to a BioGel P-30 column to remove the access dye. The labeled viruses were retrieved into the provided collection tubes in the elution buffer, and then concentrated using ultra-centrifugal filter tubes (Millipore) to reach a final concentration of about 1.0 × 10^12^ v.g./mL. 

HeLa cells pretransfected with an AAVR overexpression plasmid were seeded onto glass coverslips in 24-well plates at a density of 5.0 × 10^4^ per well. Cells were incubated with Alexa568-labeled AAV2-Luc (m.o.i. of 5000 v.g./cell) on ice for 1 h, then shifted to 37 °C in culture for 30 min for chasing the endocytosis events. Fixation was performed in 4% paraformaldehyde at room temperature for 30 min. The fixed cells were treated with a buffer containing 0.3% Triton X-100 and 2% bovine serum albumin for 30 min and probed with 1:500 diluted monoclonal antibody against anti-KIAA0319L (Abcam, Cambridge, UK). Alexa Fluor® 488-labeled donkey anti-mouse IgG (Life Technologies, MA, USA) at 1:1000 dilution was then used for incubation for 1 h in the dark. Cells were stained with probes for trans-Golgi marker TGN46 (Novusbio, Centennial, CO, USA) and Hoechst 33258 (Sigma-Aldrich, St. Louis, Missouri, USA) for nuclei visualization. Images were acquired with a confocal system (Leica Microsystems LAS AF-TCS SP8 STED, Wetzlar, Germany). 

### 4.7. AAV Infection and Luciferase Reporter Assays 

Recombinant AAV vectors carrying a luciferase expression cassette were used to evaluate the viral transduction efficiency. HeLa cells were seeded in 24-well plates at 80% confluence (about 5 × 10^5^ cells per well). Purified AAV-Luc was used to infect the cells at desired m.o.i. for 24 h. The luciferase activities were measured using a luciferase assay system according to the manufacturer’s protocol (Promega, Madison, WI, USA). A luminometer of MiNiCHEMI (Sagecreation, Beijing, China) was used for the quantitative detection of the luminescence.

### 4.8. Surface Plasmon Resonance

SPR (Surface plasmon resonance) analyses were carried out using BIAcore T200 (GE Healthcare) equipment with a flow rate of 30 μL/min at 25 °C using 100 mM NaCl in HEPES buffer (10 mM, pH 7.4). The tested AAV1 particles suspended in sodium acetate buffer (pH 4.5) were immobilized on a CM5 sensor chip by amide coupling. The analytes of purified recombinant AAVR fusion proteins were flowed through the chip at different concentrations. The BIA evaluation software (GE Healthcare) was used for the binding affinity determination and the generation of curves. 

### 4.9. Statistical Analysis

The data were presented as means from measurements of each condition. The charts were produced with Prism 6 (GraphPad Software Inc., La Jolla, CA, USA). The statistical significance was determined by Student’s *t*-test with *p*-values lower than 0.05 being considered to be significant.

## 5. Conclusions

A sensitive and specific assay (VIRELISA) was developed for fast and reliable titering of AAV particles using recombinant high-affinity receptor proteins. VIRELISA performs with high sensitivity of LOD 5 × 10^4^ v.g. and wide linear range of 1 × 10^5^ to 5 × 10^9^ v.g. The assay is low-cost, scalable, and can be carried out within 4 h for 96 samples in a single batch. The protocol is simple and does not involve or produce biohazardous or radioactive materials or waste. As VIRELISA was demonstrated to be able to detect different serotypes AAVs with quantitative capabilities, it may be considered a generic standard in kit format for titering AAV particles for applications involving comparative assessment of multiple AAV vectors. At the very least, VIRELISA will greatly facilitate the titration and quality control of AAV purification from the crude lysate. From the development of VIRELISA, the rational engineering and proper engagement of high-affinity virus receptors was sufficiently demonstrated, and this could exemplify a common approach for the design and implementation of similar protocols.

## Figures and Tables

**Figure 1 molecules-24-03973-f001:**
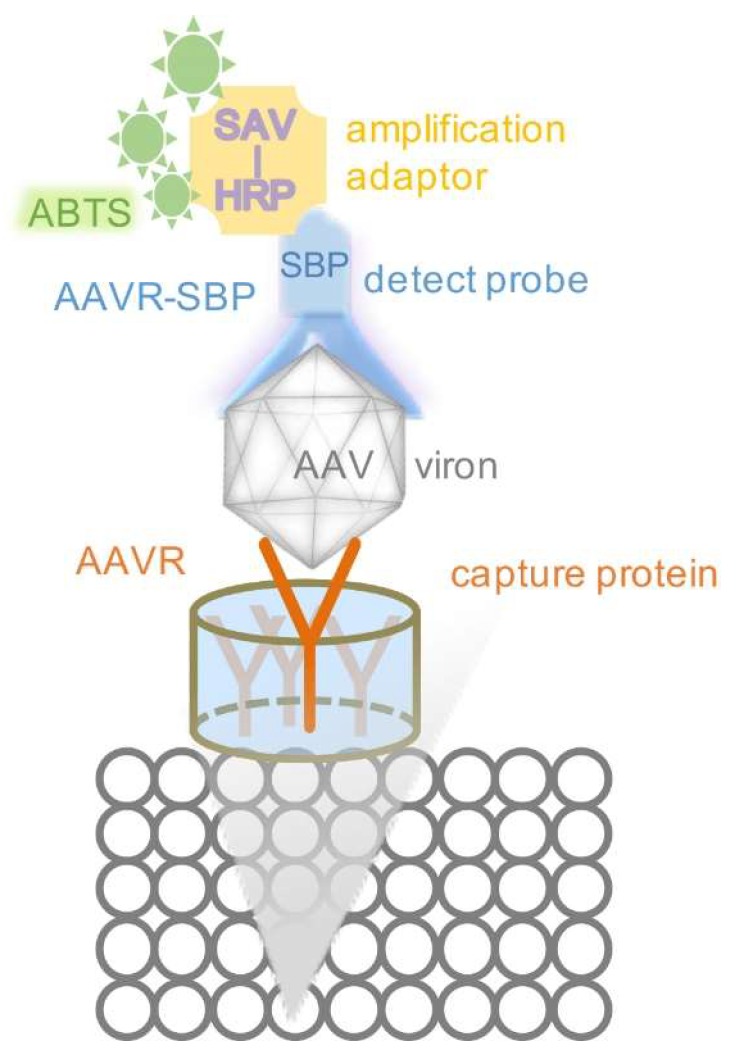
The principle of VIRELISA for schematic demonstration. The soluble AAVR was immobilized in the 96-well plate as the captured protein. AAV samples were loaded into each well. AAVR-SBP was used as the detection probe, equivalent to a primary antibody. Then SAV (streptavidin) conjugated with HRP (horseradish peroxidase) was applied as the amplification adaptor, as a secondary antibody. The ABTS (2,2′-azino-di-(3-ethylbenzthiazoline sulfonic acid)) was finally added to develop visual signals.

**Figure 2 molecules-24-03973-f002:**
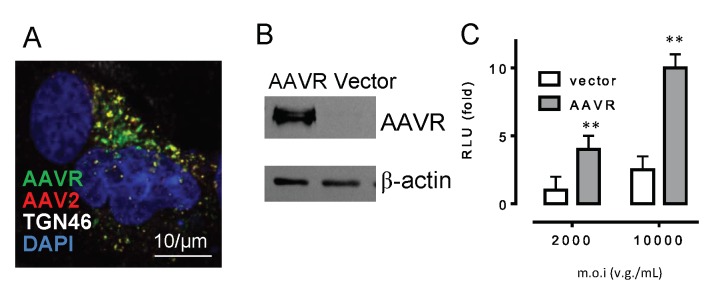
AAV2 bound to its high-affinity receptor AAVR enhanced viral transduction. (**A**) AAVR (green) concentrated in the trans-Golgi regions (white) and colocalized with AAV2 (red) in HeLa cells. (**B**) Western blot of the AAVR in HeLa cells transfected with an AAVR overexpression plasmid. (**C**) The transduction of AAV2–luciferase (m.o.i = 2000 and 10,000 v.g./cell) was measured at 24 h post-infection in both control and AAVR-overexpressed HeLa cells (** *p* < 0.01, *n* = 4).

**Figure 3 molecules-24-03973-f003:**
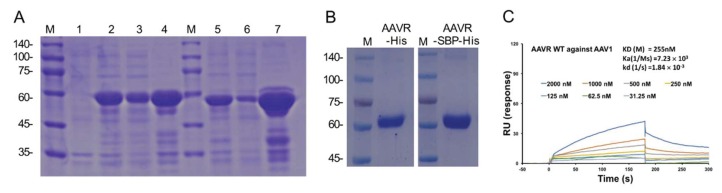
Preparation and characterization of recombinant AAVR proteins. (**A**) AAVR-His and AAVR-SBP-His proteins were expressed in *E.coli.* BL21 cells under IPTG induction. (lane 1, lysate from cells without IPTG treatments; 2, lysate of His-tagged AAVR (PKD1-5); 3, lysate supernatant from induced BL21; 4, pellet from AAVR-His expressing BL21; 5, lysate from AAVR-SBP-His expressing cells; 6, supernatant from AAVR-SBP-His positive BL21; 7, pellets from AAVR-SBP-His expressed cells. (**B**) AAVR-His or AAVR-SBP-His protein purified by nickel affinity chromatography and additional gel filtration chromatography. (**C**) Purified AAVR recombinant protein bound to AAV virion with high affinity. Soluble recombinant AAVR was tested for the binding to AAV1 capsid by BIAcore sensorgrams. The concentrations of the analytes in triplicates are indicated with the calculated association rate constant (*K*a), and dissociation rate constant (*K*d) and *K*D values.

**Figure 4 molecules-24-03973-f004:**
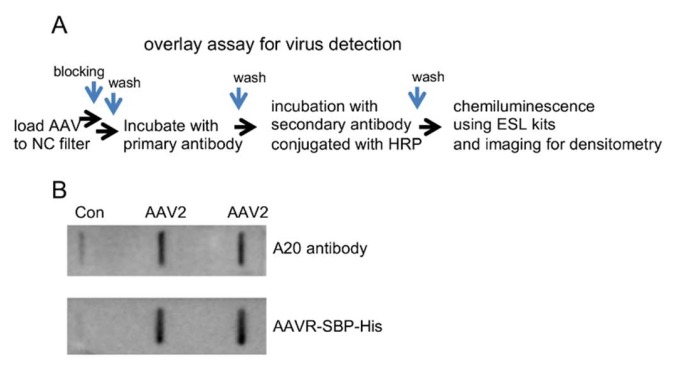
AAVR-SBP-His proteins used for viral overlay assays. (**A**) Assay workflow. AAV particles were loaded to nitrocellulose (NC) filter, probed with AAVR-SBP-His protein or A20 antibody overnight, and then incubated with HRP conjugated SAV or IgG-HRP. (**B**) AAV2 of 1 × 10^9^ v.g. subjected to the detection procedure as described in (**A**). PBST buffer was used as a negative control.

**Table 1 molecules-24-03973-t001:** Optimization of assay conditions for VIRELISA.

		AAV2 (450 nm)	Buffer (450 nm)	P/N
AAVR-SBP-His (μg/mL)	50	0.662 ± 0.003	0.192	3.4
	100	0.861 ± 0.003	0.372	2.3
SAV-HRP dilution	1/1000	0.872 ± 0.006	0.598	1.5
	1/2000	0.798 ± 0.006	0.231	3.5
	1/5000	0.662 ± 0.001	0.171	3.9
ABST incubating time (min)	10	0.320 ± 0.005	0.12	2.7
	15	0.663 ± 0.007	0.19	3.5
	20	0.819 ± 0.002	0.25	3.3
	30	1.326 ± 0.060	0.42	3.1

**Table 2 molecules-24-03973-t002:** Intra-assay and inter-assay precision of VIRELISA.

Intra-Assay		Inter-Assay	
Titer (×10^7^ v.g.)	RSD (%)	Titer (×10^7^ v.g.)	RSD (%)
10.30 ± 0.631	3.51	13.50 ± 0.605	4.47
1.32 ± 0.052	3.92	1.41 ± 0.065	4.59
0.12 ± 0.004	4.05	0.13 ± 0.005	3.92

**Table 3 molecules-24-03973-t003:** Analytical Recovery determined by adding purified AAV2 to crud lysate.

Original (×10^7^ v.g.)	Spiked (×10^7^ v.g.)	Expected (×10^7^ v.g.)	Measured (×10^7^ v.g.)	Recovery (%)
0.10	10.00	10.10	10.21 ± 0.721	108.4
	1.00	1.10	1.16 ± 0.072	105.5
	0.10	0.20	0.20 ± 0.002	102.2

**Table 4 molecules-24-03973-t004:** Selectivity of VIRELISA.

Interfering Substrate	Concentration	Standard (×10^8^ v.g.)	Spike-in (×10^7^ v.g.)	Recovery (%)
BSA (μg/μL)	10	1.0	1.028	102.8
	20		1.127	112.7
boiled AAV2 (v.g.)	1.0 × 10^6^		1.127	112.7
	1.0 × 10^7^		1.156	115.6
DEO (%)	0.05		0.988	98.9
	0.5		1.015	101.5
CsCl (g/cm^3^)	0.015		9.885	98.9
	0.15		9.630	96.3

**Table 5 molecules-24-03973-t005:** VIRELISA for AAV1 in crude lysate and comparison with qPCR.

VIRELISA			qPCR		
Titer (×10^7^v.g.)	RSD (%)	Recovery (%)	Ttiter (×10^7^v.g.)	RSD (%)	Recovery (%)
10.55 ± 0.520	4.96	105.5	10.41 ± 0.20	1.93	104.1
1.03 ± 0.047	4.47	103.2	1.06 ± 0.01	1.17	103.7

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
