# Peer review of "A Generic Method for Fast and Sensitive Detection of Adeno-Associated Viruses Using Modified AAV Receptor Recombinant Proteins"

_molecules, 2019, doi:10.3390/molecules24213973_

Round 1
Reviewer 1 Report
In this manuscript, Cui M et al established an ELISA assay based on the binding of the PKD1-5 of the AAV receptor (AAVR; KIAA0319L) with AAV virions. They demonstrated that the VIRELISA worked in quantification of AAVs, at least AAV1 and AAV2, in a range of from 10^5 to 10^9. The VIAELISA is innovative in that it could quantify AAV variants that do not have an anti-capsid antibody developed. So far, only the AAV4-lineaged AAVs do not bind AAVR, therefore the assay could have a broad application in the future. They determined the Kd of the AAVR with AAV1, which is similar to that reported for AAV2/AAVR. However, it is likely that different AAVs have various affinities with AAVR, which should be considered in quantification of different AAV variants.
Minor points:
Overall, the figure legends were described too simple, for examples, Fig. 1., Fig. 2., Fig. 3C, and Fig. 4. Please add more details. “AAV4 does not bind AAVR” should also refer the paper in J Virol. 2018 Mar 14;92(7). pii: e02213-17.Author Response
Comments and Suggestions for Authors:
In this manuscript, Cui M et al established an ELISA assay based on the binding of the PKD1-5 of the AAV receptor (AAVR; KIAA0319L) with AAV virions. They demonstrated that the VIRELISA worked in quantification of AAVs, at least AAV1 and AAV2, in a range of from 10^5 to 10^9. The VIAELISA is innovative in that it could quantify AAV variants that do not have an anti-capsid antibody developed. So far, only the AAV4-lineaged AAVs do not bind AAVR, therefore the assay could have a broad application in the future. They determined the Kd of the AAVR with AAV1, which is similar to that reported for AAV2/AAVR. However, it is likely that different AAVs have various affinities with AAVR, which should be considered in quantification of different AAV variants.
Response: Thanks for the suggestion of reviewer. Yes, indeed, different AAVs do have different affinities with AAVR as we examined, thus a standard curve needs to be prepared for the titration analysis from serotype to serotype. A potential solution to improve VIRELISA application is to find a type “standard virion” which can be used for as many as possible serotypes, or, by adjust the AAV2 curve with a weight factor so as to adapt different serotype titering. With our current data, the binding constant alone of the serotype available appeared insufficient to derive such factors.
Minor points:
Overall, the figure legends were described too simple, for examples, Fig. 1., Fig. 2., Fig. 3C, and Fig. 4. Please add more details. “AAV4 does not bind AAVR” should also refer the paper in J Virol. 2018 Mar 14;92(7). pii: e02213-17.
Response: We have expanded the figure legend with more detailed descriptions as shown in Fig.1 (page 5).Fig.2 (page 6), Fig.3 and Fig.4 (page 7). Thanks for your important suggestion, we cited the reference on page 4 line 9.
Reviewer 2 Report
The assay described in this manuscript would be off great general interest but further experimental data must be aquired to validate the findings.
1) First the assay is characterized as "universal" for all AAV serotypes when only 2 are being tested in the current work. It is critical to include other serotypes.
2) there are currently 3 commercially available kit to measure total capsids for AAV2, 8 and 9. It is critical that the assay developped in this manuscript present data that can be directly bridged with the commercial ELISA.
The poor quality of the english writting makes it very hard to read the manuscript which requires a lot of editing. The hypotheses and experimental designs are otherwise clearly explained.
Author Response
Comments and Suggestions for Authors
The assay described in this manuscript would be off great general interest but further experimental data must be aquired to validate the findings.
First the assay is characterized as "universal" for all AAV serotypes when only 2 are being tested in the current work. It is critical to include other serotypes.
Response:In the present study, VIRELISA was better characterized in the application in AAV2, since a robust commercial antibody A20 can be used for cross referencing. In our related research projects, the binding affinity of AAVR to other AAV serotypes was obtained from BiAcore assay, e.g., AAV1 and AAV5 (Nat Commun. 2019, 10(1):3760). These have provided the bases to expand the VIRELISA method for the detection of other AAV serotypes, especially for AAV1 and AAV5 which we prioritized. We took the reviewer’s critique seriously by changing the description of VIRELISA from “universal” to “genetic”. As shown in the attached figure, the results from overlay assay using AAVR-SRP-His for detecting both AAV1 and AAV5 were provided.
Detection of AAV1 and AAV5 particles as semiquantifiable blots. AAV1 or AAV5 (1 × 10^9 v.g. as titered by qPCR) was loaded to nitrocellulose filters and probed by soluble AAVR protein. PBST was used as the control (CON).
2) there are currently 3 commercially available kit to measure total capsids for AAV2, 8 and 9. It is critical that the assay developed in this manuscript present data that can be directly bridged with the commercial ELISA.
Response:We recognized that there are several commercial available kits for detecting different AAV serotypes, some of them are ELISA based, such as the ELISA kit which Grimm D, et al., reported in Gene Ther.1999, 6(7):1322-30.With our current data, the VIRELISA demonstrated a comparative performance and a wider detecting range of 1×10^5 to 5×10^9 v.g.. Except for the A20 based kit, we have not yet tested others for exact comparison, but it will sure be the future task to do. We think the immediate focus will be first validate VIRELISA for its robustness in different laboratory tests, and then develop a recombinant chimeric virion for producing a universal standard curve for different serotypes.
The poor quality of the english writting makes it very hard to read the manuscript which requires a lot of editing. The hypotheses and experimental designs are otherwise clearly explained.
Response:We apologize for the inconvenience from the poor language in writing. We have asked a native English speaker to help us for the revision and proofread of the manuscript.
Reviewer 3 Report
Cui et al. report a new sandwich ELISA-like method for titering AAV capsid using purified AAVR proteins, termed virus receptor-linked immunosorbent assay (VIRELISA). In contrast to standard ELISAs, the method reported in this manuscript does not rely on antigen-antibody reactions but instead takes advantage of virus particle-virus receptor interactions. The authors demonstrate that the dynamic range of the VIRELISA for AAV2 capsid titering is comparable to the conventional A20 antibody-based AAV2 capsid ELISA, and show its potential utility in determining capsid titers of other AAV serotypes. Such a new method that can determine AAV capsid titers is useful in the research community. However, major weaknesses are also identified. The major weaknesses are: the lack of clarify in the His-tagged AAVR and His-tagged AAVR-SBP constructs; the lack of rigor in editing the manuscript; and the use of an inappropriate approach to validating the VIRELISA using an uncharacterized AAV vector preparation. To improve the manuscript, the authors will need to address the following critiques.
Major critiques:
The authors describe in the Materials and Methods section "The AAVR complementary DNA encoding PKD domains 1-5 was fused at the C-terminus with 6× His-tag or streptavidin binding peptide (SBP) and 6× His-tag in tandem and cloned into a pET-28a vector." However, this information is not sufficient for readers to reproduce the AAVR constructs created and tested in the manuscript. The authors need to clearly describe the structures of the AAVR constructs including DNA and/or amino acid sequence information, and other associated information, such as PCR primer sequences if used, so that the readers can reproduce the reagents the authors used.
There are numerous careless errors and grammatical errors in the manuscript that have been overlooked. In addition, there are many sentences that are difficult to comprehend and need to be fixed. The following are just a few examples of many. The authors need to carefully edit and proofread the manuscript before re-submitting the manuscript for review.
Line 46: were start... Line 150: samples duplicates... Line 150: As is showed... Line 152: significant improved... Line 160: was in linear range... Line 165: the robust and ascertain... Line 172: first random selected... Line 202: AAV subtype... Line 204: able to directly indicating... Line 230: which frequently compromised to... Line 236: can be potentially using for... Line 239: may be also tried for evaluate.... Line 248: underdevelopment (as opposed to under development) Line 249: chimeras virons... (note: to be correct, chimeric virions). Line 250: can be also capable for detecting... Line 254: triple-plasmid transfection (a meaningless hyphen). Line 258: cells were co-transfected plasmids... Line 292: dilutions was loaded... Line 375: the assay is low costs... Line 376: the protocol is simple and do not involve... "virons" are found in many places. They should be "virions".
Their approach to validate the VIRELISA using an uncharacterized AAV vector-containing sample is not valid (Table 5). Unless the information about the full-to-empty AAV capsid ratio of the sample is provided, the authors would not be able to validate the assay. It is most likely that VIRELISA titers (i.e., AAV capsid titers including empty capsids) would be much higher than qPCR titers (i.e., titers of genome-packaged AAV capsids only) if both VIRELISA and qPCR assays were accurate because there are most likely abundant empty capsids in non-purified AAV vector crude cell lysate preparations. Thus, the data present in Table 5 most likely suggests that VIRELISA underestimates the viral titers.
Minor critiques:
Acronyms often appear without clear definitions. Acronyms should be fully spelled out when they first appear in the manuscript. The following are examples: SAV, LOD, LOQ, RSD, and DEO.
Figure 2. ** is missing in the figure.
Figure 3. In Panel C, the authors need to describe what assay they used to obtain the data presented (i.e., surface plasmon resonance). This important information is missing.
"A20 is only applicable for the serotype of AAV2": The authors need to appreciate that A20 can also recognize the AAV3 capsid.
Lines 239-240: It is not clear how VIRELISA can be employed to evaluate wild-type AAV infection in individuals. Do the authors think that AAV particles are continually produced in the body of infected individuals?
Line 250: "The VIRELISA method is also capable of detecting chimeric virions": No evidence for this statement is provided.
Line 299: Does the HRP-conjugated goat anti-bovine IgG the authors used cross-react with mouse IgG? If "anti-bovine IgG" is an error, please correct this sentence and provide correct information.
Line 325: "three duplicates of each sample" is confusing. Is this "triplicated samples"?
Conflict of Interest: Patent filing could be considered as COI. The authors are encouraged to contact the journal's office regarding COI.
Author Response
Comments and Suggestions for Authors
Cui et al. report a new sandwich ELISA-like method for titering AAV capsid using purified AAVR proteins, termed virus receptor-linked immunosorbent assay (VIRELISA). In contrast to standard ELISAs, the method reported in this manuscript does not rely on antigen-antibody reactions but instead takes advantage of virus particle-virus receptor interactions. The authors demonstrate that the dynamic range of the VIRELISA for AAV2 capsid titering is comparable to the conventional A20 antibody-based AAV2 capsid ELISA, and show its potential utility in determining capsid titers of other AAV serotypes. Such a new method that can determine AAV capsid titers is useful in the research community. However, major weaknesses are also identified. The major weaknesses are: the lack of clarify in the His-tagged AAVR and His-tagged AAVR-SBP constructs; the lack of rigor in editing the manuscript; and the use of an inappropriate approach to validating the VIRELISA using an uncharacterized AAV vector preparation. To improve the manuscript, the authors will need to address the following critiques.
Major critiques:
The authors describe in the Materials and Methods section "The AAVR complementary DNA encoding PKD domains 1-5 was fused at the C-terminus with 6× His-tag or streptavidin binding peptide (SBP) and 6× His-tag in tandem and cloned into a pET-28a vector." However, this information is not sufficient for readers to reproduce the AAVR constructs created and tested in the manuscript. The authors need to clearly describe the structures of the AAVR constructs including DNA and/or amino acid sequence information, and other associated information, such as PCR primer sequences if used, so that the readers can reproduce the reagents the authors used.
Response: The information about the amino acid sequence of AAVR PKD domains 1-5 and SBP-tag protein and together with the primers for preparing the PET28a-AAVR-SBP-His plasmid were supplied in the supplementary materials. The construction details had been added in manuscript (page 12, line 360-363).
There are numerous careless errors and grammatical errors in the manuscript that have been overlooked. In addition, there are many sentences that are difficult to comprehend and need to be fixed. The following are just a few examples of many. The authors need to carefully edit and proofread the manuscript before re-submitting the manuscript for review.
Line 46: were start... Line 150: samples duplicates... Line 150: As is showed... Line 152: significant improved... Line 160: was in linear range... Line 165: the robust and ascertain... Line 172: first random selected... Line 202: AAV subtype... Line 204: able to directly indicating... Line 230: which frequently compromised to... Line 236: can be potentially using for... Line 239: may be also tried for evaluate.... Line 248: underdevelopment (as opposed to under development) Line 249: chimeras virons. (note: to be correct, chimeric virions). Line 250: can be also capable for detecting... Line 254: triple-plasmid transfection (a meaningless hyphen). Line 258: cells were co-transfected plasmids... Line 292: dilutions was loaded... Line 375: the assay is low costs... Line 376: the protocol is simple and do not involve... "virons" are found in many places. They should be "virions".
Response: Thank you indeed for your careful reading and pointing out the writing errors. We revised the text together with an English native speaker and corrected the language mistakes including the listed ones, as marked in red color in the manuscript.
Their approach to validate the VIRELISA using an uncharacterized AAV vector-containing sample is not valid (Table 5). Unless the information about the full-to-empty AAV capsid ratio of the sample is provided, the authors would not be able to validate the assay. It is most likely that VIRELISA titers (i.e., AAV capsid titers including empty capsids) would be much higher than qPCR titers (i.e., titers of genome-packaged AAV capsids only) if both VIRELISA and qPCR assays were accurate because there are most likely abundant empty capsids in non-purified AAV vector crude cell lysate preparations. Thus, the data present in Table 5 most likely suggests that VIRELISA underestimates the viral titers.
Response: We agree with the reviewer that the existence of empty AAV capsids will be a major factor to influence the accuracy of VIRELISA. As the functional titer is defined by the transduction rate, PCR base methods are likely to give more correlated measures. We did experienced batch-to-batch variations in VIRELISA in terms of the correlation with the qPCR titering, however, we were not able to explain the observation based on the full/empty ratios, since even the calculation from negative staining EM and cryoEM sometimes gave inconsistent results. At least, within the same batch of viral preparation, our titering by our methods was demonstrated to be accurate and robust.
One of our suggested primary application of VIRELISA was to detect intact AAV particles in the crude lysate for the estimation regarding the yield and quality of the preparation prior to the purification procedures. As we demonstrated for AAV1 applications. In this case, besides the underestimation of viral capsids by VIRILISA, it can not complete the possibility of inaccurate determination by qPCR. Another remained possibility is that the fusion of SBP fragment somehow affected the binding of AAVR to AAV virions at a dose-dependent manner. Since we did not directly compare the binding of AAVR-SBP to AAVR alone, we do not the data regarding their differences in binding constants.
Basically, VIRELISA and qPCR methods are designed based on different principles. At this point we can admit that qPCR method could be less influenced by the empty capsid problem, nonetheless, we did not intend to use VIRELISA to reflect the empty capsid ratio before additional robust tests are performed.
Minor critiques:
Acronyms often appear without clear definitions. Acronyms should be fully spelled out when they first appear in the manuscript. The following are examples: SAV, LOD, LOQ, RSD, and DEO.
Response: We checked all the abbreviations and made necessary amendments as recommended.
Figure 2. ** is missing in the figure.
Response: Correction has been made, see in the figure2, page5.
Figure 3. In Panel C, the authors need to describe what assay they used to obtain the data presented (i.e., surface plasmon resonance). This important information is missing.
Response: The more details had been supplemented in figure 3C. in page6.
"A20 is only applicable for the serotype of AAV2": The authors need to appreciate that A20 can also recognize the AAV3 capsid.
Response: Thank you very much for pointing out the incorrect statement. We have modified the description about the A20 antibody in the manuscript (page 9, line 261).
Lines 239-240: It is not clear how VIRELISA can be employed to evaluate wild-type AAV infection in individuals. Do the authors think that AAV particles are continually produced in the body of infected individuals?
Response: It was a suggestion that VIRELISA could be potentially used for detect wild type AAV containing biosamples (e.g. concentrated tissue fluid or tissue extract followed by precipitation). Since we do not have direct laboratory evidence to support the idea, we removed the sentence from the text. No, we do not think infected wild type AAV can replicate by itself in tissues under normal situation.
Line 250: "The VIRELISA method is also capable of detecting chimeric virions": No evidence for this statement is provided.
Response:We apologize with this overstatement regard VIRELISA application. What we really meant to say was that in certain strategy of making chimeric AAV capsid, like the four plasmid mixing method, AAVR is perceivable to binding the produced virons in theory, therefore could be detected (qualitatively though) using our method. We modified the sentence to better explain this with specific emphasis.
Line 299: Does the HRP-conjugated goat anti-bovine IgG the authors used cross-react with mouse IgG? If "anti-bovine IgG" is an error, please correct this sentence and provide correct information.
Response:Thank you for the comment. It was indeed a typo mistake. We have corrected the “bovine” to “mouse” in the manuscript (page 12, line 383).
Line 325: "three duplicates of each sample" is confusing. Is this "triplicated samples"?
Response:Yes, “triplicated samples” is exactly what we meant. Correction was made (page 12, line 355).
Conflict of Interest: Patent filing could be considered as COI. The authors are encouraged to contact the journal's office regarding COI.
Response:
Our employer institute was the only ownership of the patent. The submission of the manuscript has received the approval of the institute. We consulted with the IP office of the research management department again, they do not see a problem with this issue, unless there is a specific form from the journal editorial office they need to sign. We will contact the journal office for COI handlings, and respond as instructed. Thanks for commenting on this issue.
Round 2
Reviewer 2 Report
Minor comments:
The authors improved the manuscript based on the reviewers' suggestion. However there is still a lot of grammatical errors and english language must be improved prior to publication.
The authors changed the description "universal" to genetic. It is unclear how this assay can be designed as genetic since it is not based on genes. This should be changed.
Major comments: If other serotypes cannot be tested at this time, at least the manuscript would gain strength and significance if the authors showed capsid titers obtained with the AAV2 capsid elisa that is commercially available. It is is simple, fast and would provide more interest for the field to further look into this assay.
Author Response
Comments and Suggestions for Authors
Minor comments:
The authors improved the manuscript based on the reviewers' suggestion. However there is still a lot of grammatical errors and english language must be improved prior to publication.
Response: Thank you for the comment. The manuscript has been edited and proofread word by word. The corrections and modifications have been marked in red as shown in the manuscript.
The authors changed the description "universal" to genetic. It is unclear how this assay can be designed as genetic since it is not based on genes. This should be changed.
Response: We apologize for the careless typo mistake. We have corrected the “genetic” to “generic”, which we actually meant, in the manuscript.
Major comments:
If other serotypes cannot be tested at this time, at least the manuscript would gain strength and significance if the authors showed capsid titers obtained with the AAV2 capsid elisa that is commercially available. It is is simple, fast and would provide more interest for the field to further look into this assay.
Response: Thank you very much for the good suggestion, which we think it is absolutely a fair request. In fact, we have no reason to against a direct comparison of the performance of VIRELISA with commercial AAV ELISA kits, such as from Progen or Fitzgerald. However, the problem we are facing is that it can be quite agonizing to obtain these products at the moment. Currently, any viral related import product is under strict inspection at the custom, it will take months to receive, left alone with the waiting time for paperwork. Beside, the university does not have an authorized the contract dealer for the distribution of such materials. If we try to order from the third party without guarantee, the price will go up high above our budget as a small lab. As an alternative, we did try to purchase AAV particles that pre-titered by ELISA method, unfortunately we could not get any in China, since nearly all of them are quantified by PCR method. The only one promised with a delivery failed to provide a credible evidence of the method been used. Therefore, we did what we can at best for now, which was, send our A20 antibody and VIRELISA titered virus to the technicians (as listed in the acknowledgement) in department of immunology at our school and performed an ELISA assay with their standard protocol and feedback the result to us. What we learned was that at high titers (from 1E+7 to 1E+8) both method gave similar readout, whereas at lower titers (less than 1E+6), VIRELISA tended to produce higher measured values with less variation than the ELISA assay using A20 as the capture antibody. We understand this was a rough estimation without strict controls, thus did not incorporate the results as a piece of formal evidence. We are sorry for not being able to complete the request experiment, but we will definitely try to perform the comparison as soon as we can in the future.